# E-Cadherin Modulation and Inter-Cellular Trafficking in Tubular Gastric Adenocarcinoma: A High-Resolution Microscopy Pilot Study

**DOI:** 10.3390/biomedicines10020349

**Published:** 2022-02-01

**Authors:** Ilona Mihaela Liliac, Bogdan Silviu Ungureanu, Claudiu Mărgăritescu, Victor Mihai Sacerdoțianu, Adrian Săftoiu, Laurențiu Mogoantă, Emil Moraru, Daniel Pirici

**Affiliations:** 1PhD Student, Doctoral School, Department of Histology, University of Medicine and Pharmacy of Craiova, 200349 Craiova, Romania; ilona.mihaela.liliac@gmail.com; 2Department of Gastroenterology, University of Medicine and Pharmacy of Craiova, 200349 Craiova, Romania; bogdan.ungureanu@umfcv.ro (B.S.U.); sacerdotianumihai@gmail.com (V.M.S.); 3Department of Pathology, University of Medicine and Pharmacy of Craiova, 200349 Craiova, Romania; 4Department of Research Methodology, University of Medicine and Pharmacy of Craiova, 200349 Craiova, Romania; adrian.saftoiu@umfcv.ro; 5Department of Histology, University of Medicine and Pharmacy of Craiova, 200349 Craiova, Romania; laurentiu.mogoanta@umfcv.ro; 6Department of Surgery, University of Medicine and Pharmacy of Craiova, 200349 Craiova, Romania; emil.moraru@umfcv.ro

**Keywords:** gastric cancer, tubular adenocarcinoma, E-cadherin, golgi system, lysosomes, exosomes, colocalization

## Abstract

Despite the numerous advances in tumor molecular biology and chemotherapy options, gastric adenocarcinoma is still the most frequent form of gastric cancer. One of the core proteins that regulates inter-cellular adhesion, E-cadherin plays important roles in tumorigenesis as well as in tumor progression; however, the exact expression changes and modulation that occur in gastric cancer are not yet fully understood. In an attempt to estimate if the synthesis/degradation balance matches the final membrane expression of this adhesion molecule in cancer tissue, we assessed the proportion of E-cadherin that is found in the Golgi vesicles as well as in the lysosomal pathway We utilized archived tissue fragments from 18 patients with well and poorly differentiated intestinal types of gastric cancer and 5 samples of normal gastric mucosa, by using high-magnification multispectral microscopy and high-resolution fluorescence deconvolution microscopy. Our data showed that E-cadherin is not only expressed in the membrane, but also in the cytoplasm of normal and tumor gastric epithelia. E-cadherin colocalization with the Golgian vesicles seemed to be increasing with less differentiated tumors, while co-localization with the lysosomal system decreased in tumor tissue; however, the membrane expression of the adhesion molecule clearly dropped from well to poorly differentiated tumors. Thus E-cadherin seems to be more abundantly synthetized than eliminated via lysosomes/exosomes in less differentiated tumors, suggesting that post-translational modifications, such as cleavage, conformational inactivation, or exocytosis, are responsible for the net drop of E-cadherin at the level of the membrane in more anaplastic tumors. This behavior is in perfect accordance with the concept of partial epithelial-to-mesenchymal transition (P-EMT), when the E-cadherin expression of tumor cells is in fact not downregulated but redistributed away from the membrane in recycling vesicles. Moreover, our high-resolution deconvolution microscopy study showed for the first time, at the tissue level, the presence of Lysosome-associated membrane glycoprotein 1 (LAMP1)-positive exosomes/multivesicular bodies being trafficked across the membranes of tumor epithelial cells. Altogether, a myriad of putative modulatory pathways is available as a treatment turning point, even if we are to only consider the metabolism of membrane E-cadherin regulation. Future super-resolution microscopy studies are needed to clarify the extent of lysosome/exosome exchange between tumor cells and with the surrounding stroma, in histopathology samples or even in vivo.

## 1. Introduction

Tumors of the digestive tract have an immense burden on the human life, both on social and economic grounds. Despite numerous advances, molecular pathology of cancer, and especially that of the mechanisms that control tumor extension and metastasization, are still incompletely charted.

According to the American Cancer Society’s Cancer Statistics Center, stomach cancer primarily affects the elderly [1]. People are diagnosed at an average age of 68. Men (about 1 in 96) had a higher risk of having stomach cancer than women (about 1 in 152) in 2021. However, other additional factors might influence an individual’s risk. Epithelia cells of gastric mucosa attach to a common extracellular matrix substrate, as well as to neighboring cells. Cell adhesion molecules (CAMs) are the main driving force behind tissue formation and architecture [2], while adhesion abnormalities frequently arise as a result of malignancy progression and metastasis [3,4]. Cadherins, integrins, selectins, and immunoglobulins are the four main types of cell adhesion molecules. They not only physically attach cells, but also play a key role in facilitating the communication between the extracellular and intracellular environments [4]. It has been shown that adhesion strength is proportional to the quantity of E-cadherin molecules on cell surfaces [5]. As a result, epithelial E-cadherin surface levels are critical for a wide range of functions, and even small changes in E-cadherin levels have a significant effect on a variety of processes, such as cell rearrangement changes, multiplication, and tissue architecture [6,7]. Inactivation of E-cadherin in human malignancies may be classified into two broad categories: those that lead to the synthesis of a nonfunctional protein and those that lead to a complete lack of E-cadherin [8].

A critical stage in the metastatic cascade is the loss or malfunctional cell adhesion, which generally includes, but not always, downregulation of E-cadherin expression in cancer instances [9,10]. Diffuse gastric carcinoma [11,12], invasive lobular breast cancer [13], plasmacytoid urothelial carcinoma [14,15], or pseudopapillary neoplasm of the pancreas [16] are some of the tumor types known to frequently show reduced E-cadherin expression. These tumors morphologically share the loss of tumor cell cohesion. Overall, the evidence suggests that low E-cadherin expression is associated with a poor tumor prognosis to some extent, but it cannot be used as a key predictor of a more aggressive course of disease [17].

Numerous cancer-related hereditary missense genetic changes in the E-cadherin gene in individuals with germline diffuse stomach cancer disrupt the process of translation and maturation of the molecule in its pathway towards the cell surface. This reveals that genetic abnormalities in the control and regulation of the E-cadherin cell surface contribute to cancer growth. Additionally, regardless of the histopathological grading of tumors, the analysis of these genetic anomalies explains the molecular processes behind cadherin modulation at the cell surface [18].

We have aimed here to address the question of whether E-cadherin synthesis/degradation balance, as detected by immunohistochemistry on human gastric adenocarcinoma histopathological tissue, matches its end-point expression levels, depending on tumor grading.

## 2. Materials and Methods

### 2.1. Patients and Specimens

In this study, we analyzed specimens from 18 consecutive patients who underwent surgery for primary gastric cancer, performed in the Surgery Clinics of the Emergency County Hospital of Craiova, Romania, diagnosed with well/moderate and solid tubular gastric adenocarcinoma (Table 1), and without any prior chemotherapeutic treatment. Patients were diagnosed with gastric cancer in the gastroenterology/pathology departments of the Emergency County Hospital of Craiova between September 2018 and December 2018. As controls, we utilized normal gastric tissue taken from five patients who died of non-digestive pathologies. In order to obtain clinical information from the patients, we reviewed their medical records for gender, age, tumor localization, grade, extension of disease, T and N stages according to the American Joint Committee on Cancer (8th edition, 2017) [19], histological grading according to World Health Organization (5th edition, 2019) [20], criteria including well, moderately, and poorly differentiated adenocarcinoma, based on tubular formation.

Based on the initial histopathological reports, we chose the most representative tissue block from each case and re-confirmed the pathology and grading. We included only cases for which more than 80% of the tumor area reflected the respective grading. Moreover, in order to eliminate any inter-observational subjective gradings, we have grouped the cases as either (i) low-grade (tubular well and moderately differentiated) tumors or (ii) tubular (solid) poorly differentiated adenocarcinoma.

### 2.2. Tissue Processing and Immunohistochemistry

Selected paraffin-embedded archived tissue blocks were sectioned as either 2 μm-thick sections (for hematoxylin and eosin standing, as well as for enzymatic immunohistochemistry), or as 5 μm-thick sections (for fluorescence immunohistochemistry). After re-confirming the pathologies and tumor gradings, slides were processed for individual immunohistochemical detection of E-cadherin (mouse, clone NCH-38, Dako, Glostrup Denmark), endoplasmic reticulum marker Giantin (rabbit, polyclonal, ab80864, Abcam plc, Cambridge, UK), and lysosomal marker LAMP1 (Osteoblast-Derived Lysosomal Membrane Protein 1) (rabbit, polyclonal, ab24170, Abcam). Briefly, the sections were deparaffinized in xylene, rehydrated in graded alcohol series, processed for antigen retrieval by microwaving in 0.1 M citrate buffer pH6 for 20 min, incubated in 1% hydrogen peroxide in distilled water for 30 min to block the endogenous peroxidase activity, and kept for another 30 min in 3% skimmed milk in PBS for blocking unspecific antigen sites. For enzymatic single immunohistochemistry, the primary antibodies were incubated on the slides at 4 °C for 18 h (E-cadherin, 1:100; Giantin, 1:100; LAMP1, 1:300), and the next day the signal was amplified for 60 min utilizing a species-specific peroxidase polymer-based system adsorbed for human immunoglobulins (Nikirei-Bioscience, Tokyo, Japan). The signal was then detected with 3,3′-diaminobenzidine (DAB) (Nikirei-Bioscience) and the slides were coverslipped in DPX (Sigma–Aldrich, St. Louis, MO, USA) after a hematoxylin counterstaining. All slides stained for each of the primary antibodies have been processed at the same time for protocol consistency together with control slides stained either with DAB or with hematoxylin in order to obtain pure spectral signatures of the respective stains (see Section 2.3). Negative controls were obtained by omitting the primary antibodies.

For fluorescence double immunohistochemistry, the slides were processed as described above, but instead they were incubated overnight with a pair of E-cadherin (1:50)/LAMP1 (1:200) or E-cadherin (1:50)/Giantin (1:50) primary antibodies. After thorough washing, the signals were simultaneously detected with a mix of anti-mouse Alexa Fluor 594 and anti-rabbit Alexa Fluor 488 secondary antibodies (Thermo Fisher Scientific, Waltham, MA USA; 1:300, 2 h at room temperature). In all cases, the slides were counterstained with 4′,6-diamidino-2-phenylindole (DAPI) (Thermo Fisher Scientific) for 15 min, incubated for 20 s in a 0.3% Sudan Black (Sigma–Aldrich) alcoholic solution to reduce autofluorescence, washed in distillate water and coverslipped with a fluorescence anti-fading mounting medium (Vectashield, Vector Laboratories, Burlingame, CA, USA) [21].

### 2.3. Image Processing and Statistics

High-resolution imaging of the slides was performed utilizing a Nikon 90i motorized microscope (Nikon Europe BV, Amsterdam, the Netherlands) equipped with plan apochromat high numerical aperture immersion objectives (40×, NA = 0.95; 60×, NA = 1.27; and 100×, NA = 1.45), an intermediate internal lens capable of doubling the magnification of the objective, a high-resolution low-noise 16.25 Mp DS-Ri Nikon CMOS camera (7.3 × 7.3 µm pixel-size yielding images of 4.908 × 3.264 pixels), a Nuance FX multispectral camera (Perkin Elmer, Hopkinton, MA, USA), a Prior high-resolution motorized stage, LED fluorescence modules, as well as the Nikon NIS-Elements Advanced Research image analysis package version 4.30.02, on a dedicated Dell Precision Tower 7910 graphic station (2× Intel Xeon 8-Cores E5-2630 v3 2.40 GHz, 64 GB DDR4, NVIDIA (NVIDIA Corp., Santa Clara, CA, USA) Quadro M4000 8 GB GDDR5).

Initial observation of E-cadherin immuno-expression pattern was performed on enzymatically stained slides with the aid of the Nuance FX multispectral sensor. After building a spectral library from individual slides stained with either hematoxylin or DAB (as described above), we were able to efficiently unmix and characterize the membrane/cytoplasmic expression patterns (Figure 1). A final 200× magnification power was employed in this evaluation.

For semi-quantitative analysis, 10 random high-resolution images were captured from each fluorescence-stained slide from the tumor areas, and normal gastric mucosa from control patients, with the 16.25 Mp camera utilizing the 40× objective. Images were obtained by sequential scanning of each channel with specific pairs of highly selective custom-made filters in order to eliminate the cross-talk of the fluorophores and to ensure a reliable quantification for DAPI, Alexa 488, and Alexa 594 spectra (Chroma Technology Corp., Bellows Falls, VT, USA). Higher magnification exemplificative images have been captured utilizing the highest magnification objective (100×). We have captured either single optical planes or complete z-stacks through the full thickness of the sections at a 0.1 µm z-step, and all data were archived in the *.n2d Nikon proprietary format. All fluorescence image data were processed for blind deconvolution utilizing Nikon’s 2D/3D deconvolution algorithms, at 5 iterations.

As we only targeted the tumor epithelium in this analysis, stroma was manually removed for all fluorescence image sets prior to any quantification and thus not considered any further. All fluorescent signals (E-cadherin, LAMP1, and Giantin) were quantified as area, and then reported as percentage of total epithelium in the respective image. Percentages were averaged for all images in each slide (patient), and then slides from patients with either low-grade intestinal type of adenocarcinoma (G1–G2), high-grade (G3), or normal control gastric tissue were averaged for each of these three pathological groups. 

Moreover, E-cadherin/LAMP1 and E-cadherin/Giantin colocalization ratios were calculated in Image-Pro Plus AMS 7 image analysis software (Media Cybernetics, Bethesda, MD, USA), utilizing the colocalization between their respective fluorescence channels, and were reported as two-color overlapping coefficients. Continuous numerical data that resulted from this analysis were exported and plotted in Microsoft Office Excel 2010 (Microsoft Corporation, Redmond, WA, USA) and analyzed using the SPSS software (IBM SPSS Statistics, Version 20.0). In order to assess statistical differences, we used the Student’s t-test for comparing the means of two groups and one-way ANOVA (analysis of variance) with least significant difference (LSD) post hoc analysis in order to compare the means of more than two groups. Correlations were assessed using the Pearson’s correlation coefficient. Data were reported as mean ± standard deviation of the means (SD). In all cases, *p* < 0.05 was used to indicate statistical significance. Image collages were prepared and annotated utilizing CorelDRAW Graphics Suite 2018 (Corel Corporation, Ottawa, ON, Canada).

## 3. Results

The average age of our 18 patients was of 62.88 ± 11.71 years old, compared to the average age of our control patients of 69.4 ± 7.43 years old; there were 6 females and 12 males for the study group and 2 females and 3 males for the control group (Table 1). The most frequent localization was the body of the stomach (13 cases), the most frequent tumor stage was T3 (8 cases), followed by T4a (7 cases) and T2 (3 cases).

On immunohistochemistry, we have first observed that E-cadherin was not only expressed with a strict lateral and sometimes basally-located membrane-pattern, but it also exhibited granular-diffuse or vesicular-like patterns under the membrane and, in fact, through the cytoplasm (Figure 2 and Figure 3). Indeed, high-magnification spectral unmixing revealed the clear intracytoplasmic expression, in the control gastric tissue as well as in tumor areas. Even when we considered low differentiated infiltrative tumors that showed decreased E-cadherin expression, the intracytoplasmic expression pattern could be constantly identified at 200× magnification by utilizing this technique.

For high-grade tumors, the membrane expression pattern was variable but clearly decreased; however, a fine granular-vesicular staining pattern in the cytoplasm could still be identified in these cases (Appendix A). High-resolution deconvolution fluorescence imaging also confirmed that E-cadherin colocalizes with the intracytoplasmic vesicle system in both control and tumor tissue, for both LAMP1 and Giantin markers (Figure 4, Appendix A).

We next aimed to assess the co-expression of E-cadherin maturing in the Golgi vesicles, or being degraded in the lysosomal system by analyzing the semi-quantitative expression levels of E-cadherin, Giantin, and LAMP1, as well as their colocalization degrees. As we were interested in the latest stages of E-cadherin maturation prior to its integration in the cellular membrane, we did not utilize an endoplasmic reticulum marker aside from Giantin. We aimed to evaluate the epithelial contribution only, thus we have not considered stroma in any of the further analyses (Figure 5 and Figure 6).

As expected, E-cadherin seemed to be decreasing in high-grade tumors compared to low-grade tumors; however, Golgi vesicles activity and density seemed to be increasing in the tumor epithelium of high-grade tumors, even compared to the normal tissue. LAMP1 showed a clear increase in the tumor stroma in high-grade tumors (Figure 6), however unquantified images only it was not clear what its contribution would be in the epithelia.

When we compared the epithelial expression levels of E-cadherin, LAMP1, and Giantin in all cancer cases pooled together compared to the control tissue, we recorded statistically significant differences (*p* = 0.011, *p* < 0.001, *p* = 0.037) in all three cases (Figure 7A).

E-cadherin and LAMP1 exhibited lower values compared to the control tissue; however, Giantin showed a clear tendency of actually increasing in the tumor epithelium compared to the controls. We wanted to assess these differences more thoroughly, thus we divided the tumor cases as well differentiated/poor differentiated tumors and assessed the normalized epithelial expression of the three markers (Figure 7B). E-cadherin clearly decreased from low-grade to high-grade tumors, but due to the high variability of the staining, the control tissue attained a statistically significant difference only for high-grade tumors (*p* = 0.014). LAMP1 also showed the same tendency, with an even more clear-cut difference between tumor grades (*p* = 0.021 and between low-grade/high-grade tumors and the control tissue (*p* = 0.001, *p* < 0.001). As we have mentioned above, LAMP1 expression varied and seemed to be increasing in the tumor stroma, most probably due to the inflammatory infiltrate; however, its epithelial expression varied in an opposite direction. Overall, Giantin expression was found to be higher in tumor tissue, and stratification for low-grade/high-grade tumors revealed that it was in fact drastically increasing from low-grade tumors to high-grade cancer (*p* = 0.047). It can be conceived that more aggressive tumor clones would need more intense syntheses, and this most probably explains the increase in Golgi vesicle surface area in poor differentiated tumors. 

In order to assess the balance between E-cadherin levels within the Golgian vesicles and putative lysosomal degradation system, we calculated colocalization coefficients of E-cadherin with Giantin and LAMP1 (Figure 8).

Due to the dense-diffuse and vesicular expression pattern of E-cadherin already described in high-magnification images (Figure 2 and Figure 3), LAMP1 and Giantin vesicles showed a high degree of overlapping with E-cadherin (0.756 ± 0.069% and 0.797 ± 0.087%, respectively) (Figure 8A) in tumor tissue. However, these values showed a clear-cut decrease when LAMP1/Giantin colocalizations were compared with E-cadherin for the control tissue (0.877 ± 0.023% and 0.854 ± 0.020%, respectively). The difference was statistically significant, however, only for E-cadherin/LAMP1 colocalization (*p* = 0.0015). Interestingly, the values of this denominator showed wider variability for tumor areas while being more constant for the control tissue, and this fact might be explained by the polyclonality and heterogeneity of the tumor tissue cellular metabolic rates compared to the normal epithelia. Although a decrease was also present here, the difference did not attain statistical significance for Giantin/E-cadherin colocalization in tumor tissue compared to the controls (*p* = 0.266), revealing that there was a more conserved level of overlapping with the Golgi vesicle system. Together with the statistically significant decreased colocalization and the lysosomal marker, these data suggest that E-cadherin tends to be found more frequently in the Golgian compartment compared to the lysosomal compartment. 

We wanted to explore this avenue further, so we analyzed the colocalization degrees also depending on low-grade/high-grade stratification of the tumors (Figure 8B). E-cadherin colocalization with the lysosomal compartment showed an actual tendency of decrease from low-grade to high-grade tumors, although there was no statistically significant difference between these two groups (*p* = 0.265); however, both were significantly lower than the control group (*p* = 0.016, *p* = 0.002). E-cadherin colocalization with the Golgian system even seemed to be increasing with advancing tumor grade, although due to intra-group variations, no statistically significant differences could be noted, even compared to the control group (*p* = 0.108, *p* = 0.410). Altogether, although the small sample size and the higher mosaicism in the tumor metabolic rates and protein synthesis did not lead to statistically significant differences between all the groups, these data suggest that E-cadherin immunoprofiling seems to be shifting towards the Golgi compartment with more advanced tumor staging.

Lastly, although we had a small number of cases in the study group, we sought to evaluate if any of the morphological parameters would show a correlation with the age of the patients, gender, disease extension, and the number of invaded lymph nodes (Appendix A). The E-cadherin/LAMP1 overlapping coefficient showed no correlation with the age of the patients (r = −0.04, *p* = 0.88), but the E-cadherin/Giantin overlapping coefficient showed an overall weak direct correlation with the age of the patients (r = 0.232, *p* = 0.04) (Appendix A), and when we divided the cases as either low-grade or high-grade, it was interesting to see that the tendencies were in fact opposite. Thus, for low-grade tumors, the correlation was in fact inverse (r = −0.208, *p* = 0.21), but we obtained a strong direct correlation for the high-grade tumors (r = 0.779, *p* = 0.03) (Appendix A). E-cadherin, LAMP1 and Giantin expression areas showed no correlation with the number of invaded ganglia, for low-grade and high-grade tumors. We also looked at the overlapping coefficients of E-cadherin with Giantin and LAMP1, and we could not find any correlation with the number of invaded lymph ganglia (r = −0.171, *p* = 0.78; r = 0.231, *p* = 0.67) (Appendix A). 

When we evaluated the expression of the three markers, depending on the tumor stage, the data showed that LAMP1 had a tendency to decrease with advancing stage, while Giantin was clearly increasing with a more advanced stage, although none of the trends attended statistical significance, probably due to the low number of cases available here (Figure 9A). The overlapping coefficients of E-cadherin with LAMP1 and Giantin also did not show any significant variation for the three tumor stages (Figure 9B). 

Regarding gender stratification and expression areas, E-cadherin showed a significantly lower expression level in females compared to males (*p* = 0.023), while Giantin differences, despite having the same trend, did not attain statistical significance (*p* = 0.065) (Figure 9C). E-cadherin/Giantin overlapping coefficient showed significantly lower values for females compared to male patients (*p* = 0.005), while there was no variation for E-cadherin/LAMP1 overlapping coefficients (Figure 9D). Thus, the overlapping degree of the E-cadherin with the Golgi marker seemed to be reflected mostly in the difference between females and males. 

The close image evaluation also revealed an interesting phenomenon. On 0.1 µm seriate optical sections and 3D renderings, LAMP1-positive vesicle network was identified as crossing the E-cadherin-defined membrane from one cell to another (Figure 10). Seriate image follow-up of large enough vesicles clearly showed continuity from one side of the membrane to the other. 

This phenomenon could not be identified in control gastric mucosa, nor could it be followed up in high-grade carcinoma, as E-cadherin staining had a discontinuous membrane staining pattern. Additionally, this transition was not identified in any of the slides double stained for E-cadherin and Giantin. We did not aim to further quantify the incidence of the phenomenon as it was not the initial focus of this work, nor did we have the capability to fully scan slides in confocal mode with multiple optical planes. The necessary work load and further subclassification of these vesicular systems should be analyzed separately in great detail; however, it is worth mentioning that these trans-membrane LAMP-1-positive vesicles can be identified in routine formalin fixed paraffin embedded (FFPE) diagnostic tissue.

## 4. Discussion

In this proof-of-principle study, we have sought to perform an integrative analysis on E-cadherin adhesion molecule expression in gastric adenocarcinoma, in conjunction with its localization in the final secretory pathways of the Golgian system, or during its recycling in the lysosomal system. We have showed that Golgi apparatus is enhanced in gastric cancer epithelia compared to normal gastric mucosa, and that there is even a further amplification in higher grade tumors compared to low-grade cancer. E-cadherin overlapping with the Golgian system was decreased compared to control tissue but without attaining statistical significance, and with a tendency for higher values for high-grade tumors. E-cadherin overlapping with the lysosomal system, as detected with the LAMP1 marker, recorded a more prominent drop in tumor tissues, with a statistically significant difference from control gastric areas, and this drop even accentuated for higher grade tumors. Altogether, our data showed that E-cadherin is prone to be found more in the Golgian compartment compared to other re-internalizing vesicular compartments, suggesting that E-cadherin is more predominantly synthetized than re-internalized via lysosomes, especially with advancing grade of the tumor. A direct inference of this would be that E-cadherin would be more predominantly synthetized towards the membrane rather than re-internalized via lysosomes in cancer tissue, without being able to rule out more complex modifications, such as externalization through exosomes or synthesis of a non-functional molecule, that would not be able to integrate into the membrane. This is in fact not the first study to have shown cytoplasmatic E-cadherin expression and colocalization in the Golgi vesicles, as another Golgi marker, the lectin Griffonia simplicifolia II, has been previously utilized to show this phenomenon. Although these authors have not utilized spectral unmixing, nor double immunofluorescence to fully detect and semi-quantitatively assess two signal colocalization, they have shown the same high variability of the co-stainings as described in our statistical analysis [22].

The Golgi apparatus suffers morphological changes in tumors, ranging from minimal enlargement to severe fragmenting [23]. The defective glycosylation in cancer results in an increase in sialylation, which has been linked with a metastatic cell phenotype in both clinical and experimental setups. Giantin is the Golgi matrix protein with the highest molecular weight (376 kDa) and is required for the cross-bridging of cisternae during the formation of Golgi vesicular system. Changes in the Giantin structure, such as dedimerization, may contribute to the morphological dysregulation and functional N- or O-glycan sialylation increase that is typically found in cancer [23]. An overall increase in the function of the Golgian system might be expected in cancer tissue, with cells that require faster metabolism and synthesis rates; however, as E-cadherin decreases on the membranes of higher-grade cancer cells, abnormal modification and maturation most likely explain the end decrease of its membranous expression. Indeed, it has been demonstrated that O-linked β-N-acetylglucosamine (O-GlcNAc) modification delays E-cadherin processing in the endoplasmic reticulum, and incomplete processing by proprotein convertases delays E-cadherin transit later in the secretory pathways [24]. A study of exocytic transport of E-cadherin indicated that it is carried from the trans-Golgi network towards the Rab11-positive recycling endosomes in transit to the surface of the cell [25]. Considering that Giantin is necessary for cross-bridging cisternae during Golgi formation [26,27], de-dimerization of Giantin may result in Golgi structural instability. By influencing the activation of proapoptotic kinases, altering the Golgi may ensure cancer cells’ survival. On the one hand, the development of a fragmented Golgi profile is a factor of carcinogenesis; on the other hand, it may be viewed as a result of cancer progression [23]. There is no other morphological study yet detailing the expression of Giantin in the epithelial component of gastric carcinoma at the tissue level; however, as the Golgi vesicles suffer fragmentation and morphological changes in cancer cells, it is difficult to interpret if its detection by immunohistochemistry would also point towards a functional protein. Its exact function is not known, and very recently it has been proposed that contrary to the classical view that Giantin would promote trans-vesicle transport of maturing glycol-proteins, it might in fact result in delayed transport between stacks that may enable a correct glycosylation of proteins passing towards the membrane [28]. It is worth mentioning that none of the above-mentioned studies selected only epithelium for analysis, but rather performed quantitative studies on whole tissue without separating the stroma. We have also experienced a high variability of the Giantin signal, but our analysis only encompassed the stroma and the intestinal-type of gastric cancer, without considering signet-ring and mucinous adenocarcinoma [22].

On the other hand, for recycling, the E-cadherin complex is first exposed to tyrosine phosphorylation and ubiquitination before being taken up by the cell’s endocytosis process. By using a modified yeast 2-hybrid system, Hakai, an E-cadherin binding protein, was isolated and named as E3 ubiquitin-ligase [29]. Hakai binds E-cadherin in a tyrosine phosphorylation-dependent way, causing the E-cadherin complex to be ubiquitinated. In epithelial cells, the modulation of the expression of Hakai changes cell-to-cell adhesion, which promotes E-cadherin endocytosis and increases cell mobility. Hakai can thus affect cell adhesion and may play a role in the control of EMT during development or metastasis via dynamic recycling of E-cadherin [29]. Coordinated transcription and epigenetic mechanisms regulate lysosomal biogenesis, which is crucial for cancer metabolism and plays an important role in its progression [30]. 

Moreover, LAMP1 is not only expressed on the lysosomes, but also on the exosomes and multivesicular bodies, reflecting many recycling and transcytosis pathways [31,32]. The complexity of the myriad of connections that drive the fate of the proteins in the lysosome—exosome pathway is further illustrated, for example, by the fact that drug-resistant carcinoma cells are able of selectively sort lysosomal proteins into the exosomal pathway, and thus escape of the effects of chemotherapy [33,34]. Drugs that diffuse into the cytoplasm are trapped in lysosomes and are rapidly effluxed by cancer cells that express multidrug resistance transporters. Doxorubicin, cisplatin, vincristine, vinblastine, sorafenib, and sunitinib are some examples of lysosomotropic chemotherapeutics [34,35,36]. Additionally, the capacity of cancer cells to alter the microenvironment via hijacking the lysosomal exocytosis process is critical for cancer growth. By merging with plasma membrane, lysosomes discharge their soluble and particulate elements extracellularly, altering the content of plasma membrane, acidifying the tumor microenvironment, and damaging the extracellular matrix. These processes together provide the optimal environment for cancer cell movement, invasion, and tumor metastasis [36]. Because of alterations in the cytoskeletal network, lysosomal movement and exocytosis, lysosomes tend to shift at the plasma membrane as cancer progresses [37,38,39].

Interestingly, recent data showed that epithelial-to-mesenchymal transition (EMT) in most human pancreatic ductal adenocarcinoma, breast cancer, and colorectal cancer cell lines does not suppress the transcription of E-cadherin, but in fact induces its relocation from the membrane towards Rab11+ late recycling vesicles, a phenomenon called partial (P-EMT) [40]. Relocation of E-cadherin from the cell membrane led to a loss of other adhesion molecules, such as tight junction protein Claudin-7 (CLDN7) and the epithelial cell adhesion molecule (EPCAM), suggesting a pivotal role for E-cadherin in modulating the overall adhesivity of epithelial cells. The existence of P-EMT is in perfect accordance with our results, which in fact now extends these findings to gastric adenocarcinoma, and contemplates a more complex mechanism of epithelial-to-mesenchymal transition where the gain of fibroblast-like morphology and increased motility of cancer cells is followed by a loss of the epithelial phenotype but not a loss of expression for epithelial proteins. Partially retained E-cadherin expression is present during tumor cell extravasation and promotes cell cluster formation and group migration, which also confers resistance against the natural killer cells during their travel through the circulatory system. However, at the colonization site, it has been shown that metastatic tumor cells regain E-cadherin expression, which, besides increased adhesivity, also leads to a suppression of apoptotic pathways and promotes nidation and further growth. In fact, loss of E-cadherin increased invasiveness but also increased oxidative stress and apoptosis, thus reducing the overall survival and nidation capacity of metastasizing cells [41]. Together, the dynamic balance between the P-EMT and complete EMT (C-EMT) profiles seems to be a key element that drives local tumor invasiveness and distant nidation during metastasization [42].

Our high-magnification optical sectioning revealed LAMP1 vesicles as being trafficked between epithelial cancer cells, and this phenomenon could not be generalized for Giantin-positive vesicles or for normal gastric issues. Although the concept that gastric cancer cell-derived exosomes promote tumor growth and proliferation in an autocrine manner is not new [43], to our knowledge, this is the first microscopy study to have actually showed morphological evidence of the cross-membrane passage of LAMP1-positive vesicles. Exosomes have a crucial role in maintaining the tumor microenvironment [44]. They function as efficient signaling pathways between cancerous cells and the cells within the tumor microenvironment [45,46]. By transporting functional biomolecules, exosomes play an important role in gastric cancer development, including carcinogenesis, immune evasion, tumor spreading, angiogenesis, and drug resistance [47,48].

The main limitation of this pilot study is represented by the small number of patients available, without prior chemotherapeutic regimens that would have altered the metabolism of tumor cells, and from whom pathological material was available. Besides the small number of cases, the existence of P-EMT and C-EMT also explains the high variability in our measurements, at least for the morphologic denominators that show more constant values for control patients. Moreover, we have chosen tubular adenocarcinoma for our study, as this is the most common histologic type of early gastric carcinoma [49]. Along the same line, deconvolution and image processing required lengthy analysis times; it was thus our intent to carry out a proof-of-principle study that would semi-quantitatively address the issue of E-cadherin immunoexpression in different sub-cellular compartments, on routinely processed diagnostic samples at the tissue level. Although the complexity of the tumor microenvironment in human pathology cannot be entirely mimicked by in vivo studies, future functional studies on gastric adenocarcinoma cell lines are clearly needed to shed light on the connections with the cell cycle and on the functionality of E-cadherin itself through its intracytoplasmic pathways towards and from the membrane-bound isoforms.

## 5. Conclusions

Altogether, this study shows that E-cadherin trafficking and synthesis in gastric adenocarcinoma, as evidenced by immunohistochemistry on FFPE tissue, exhibits a different pattern compared to the final functionality pattern. Thus, an increased colocalization with the Golgian vesicles and a decrease in the lysosomal pathways do not translate into a higher membrane expression with increasing grade of the tumors. Post-translational modifications and complex vesicular trafficking alterations in cancer, as for example, inter-epithelial cell exosome transfer, are most likely responsible for this structural–functional decoupling of protein synthesis in cancer, and for the existence of dynamic P-EMT/C-EMT profiles during cancer progression and metastasization.

## Figures and Tables

**Figure 1 biomedicines-10-00349-f001:**
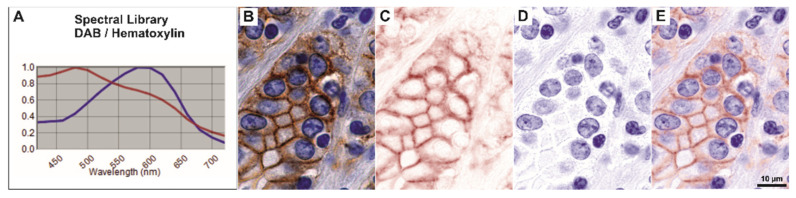
Example of spectral unmixing for enzymatic immunohistochemistry. (**A**) Spectral library built within the Nuance FX package in order to separate hematoxylin and DAB shows distinct emission peaks. (**B**) Raw RGB exemplary image of E-cadherin immunohistochemistry, as captured by the sensor in RGB mode. (**C**–**E**) Correspondent unmixed images showing separate spectra for DAB and hematoxylin, as well as the merged image, clearly show membrane and sub-membrane cytoplasmic dense-diffuse staining (**E**). Scale bar for images (**B**–**E**), 10 µm.

**Figure 2 biomedicines-10-00349-f002:**
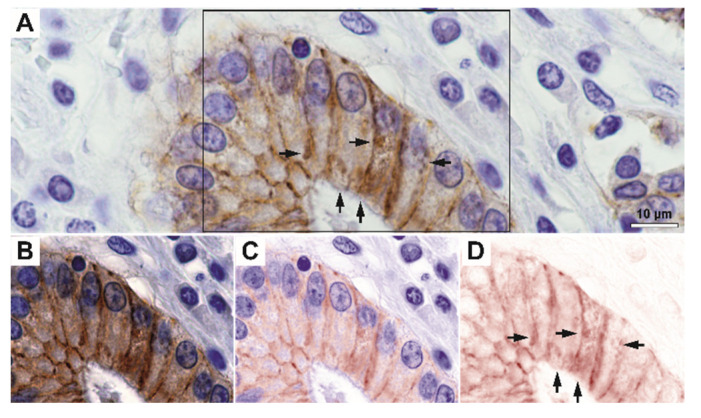
Exemplary image of control gastric glandular epithelium stained for E-cadherin. (**A**) RGB image and (**B**–**D**) unmixed individual and merged images showing membrane and intracytoplasmic staining (arrows). The rectangle in Figure (**A**) represents the selection illustrated in images (**B**–**D**). Scale bar for images (**A**–**D**), 10 µm.

**Figure 3 biomedicines-10-00349-f003:**
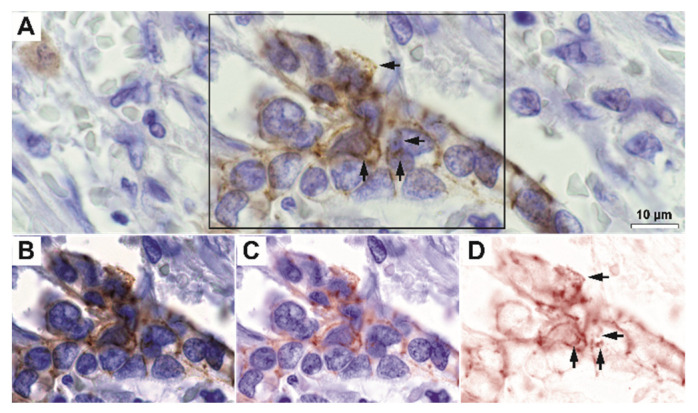
Exemplary image of solid gastric adenocarcinoma stained for E-cadherin. (**A**) RGB image and (**B**–**D**) unmixed individual and merged images showing membrane and granular and diffuse intracytoplasmic staining (arrows). The rectangle in Figure (**A**) represents the selection illustrated in images (**B**–**D**). Scale bar for images (**A**–**D**), 10 µm.

**Figure 4 biomedicines-10-00349-f004:**
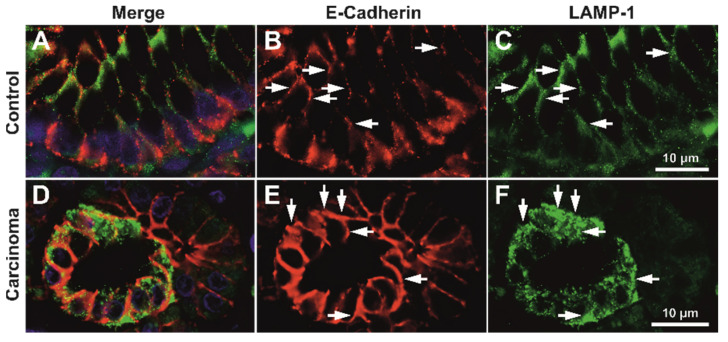
Exemplary deconvoluted images of double immunofluorescence for E-cadherin and LAMP1. On both control (**A**–**C**) and adenocarcinoma (**D**–**F**), images show colocalization of the two markers at both membrane and cytoplasmic levels. Colocalizations are indicated (arrows) at both membrane and sub-membrane cytoplasmic levels. Scale bar for images (**A**–**F**), 10 µm.

**Figure 5 biomedicines-10-00349-f005:**
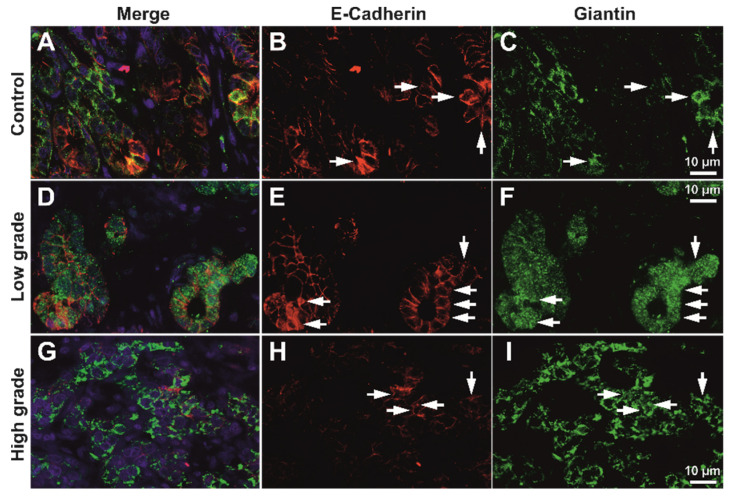
Exemplary deconvolved working images of double immunofluorescence for E-cadherin and Giantin. Epithelia of control tissue (**A**–**C**); low-grade (**D**–**F**) and high-grade (**G**–**I**) adenocarcinoma have been analyzed for expression area and Giantin/E-cadherin overlap. Colocalizations are indicated (arrows) at both membrane and sub-membrane cytoplasmic levels. Scale bar for images (**A**–**I**), 10 µm.

**Figure 6 biomedicines-10-00349-f006:**
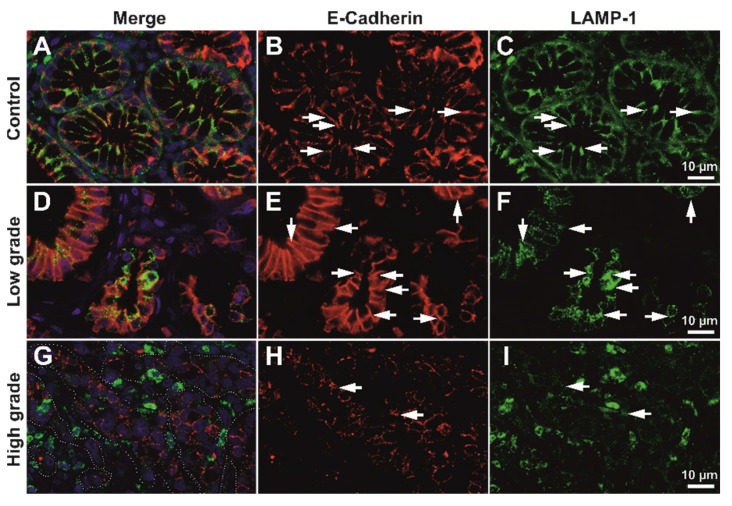
Exemplary deconvolved working images of double immunofluorescence for E-cadherin and LAMP1. Epithelia of control tissue (**A**–**C**), low-grade (**D**–**F**) and high-grade (**G**–**I**) adenocarcinoma were analyzed for expression area and LAMP1/E-cadherin overlap. In image G the dotted line illustrates how epithelia was selected as the region of interest for further analysis; colocalizations are indicated (arrows) at both membrane and sub-membrane and cytoplasmic levels. Scale bar for images (**A**–**I**), 10 µm.

**Figure 7 biomedicines-10-00349-f007:**
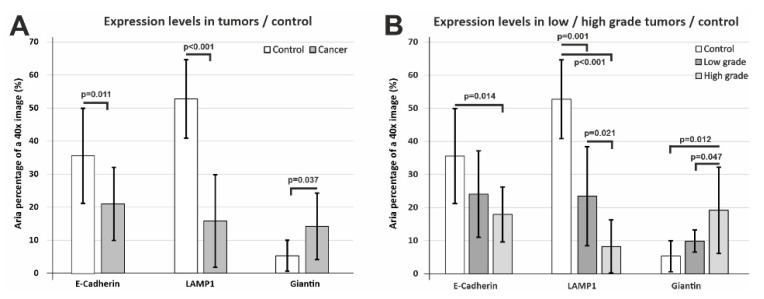
Area expression levels of E-cadherin, LAMP1 and Giantin in control and tumor epithelia. (**A**) E-cadherin and LAMP1 show overall significant decrease compared to control tissue, while Giantin exhibits a significant increase in tumor epithelia compared to the control. (**B**) Further stratification into low-grade/high-grade tumors showed that E-cadherin and LAMP1 decreased with increasing tumor anaplasia, while Giantin showed a massive increase for high-grade tumors. Significance is shown for Student’s t-test (**A**) or using a one-way ANOVA followed by a post hoc Fisher’s LSD test (**B**). Data are expressed as means ± SD.

**Figure 8 biomedicines-10-00349-f008:**
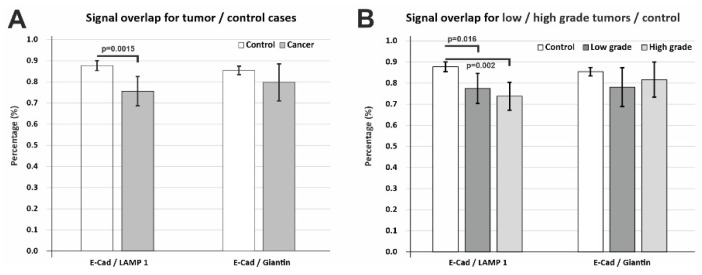
Fluorescence signal overlapping coefficients for LAMP1/E-cadherin and Giantin/E-cadherin. (**A**) Overall analysis of tumor versus control epithelia showed that the colocalization decreased in cancer, but less drastically for colocalization with Giantin. (**B**) Stratification of tumor cases into low-grade/high-grade tumors showed that colocalization with LAMP1 decreases towards less differentiated tumors, while colocalization with Giantin actually increases (although non-significantly) towards high-grade tumors. Significance is shown for Student’s t-test (**A**) or using a one-way ANOVA followed by a post hoc Fisher’s LSD test (**B**). Data are expressed as means ± SD.

**Figure 9 biomedicines-10-00349-f009:**
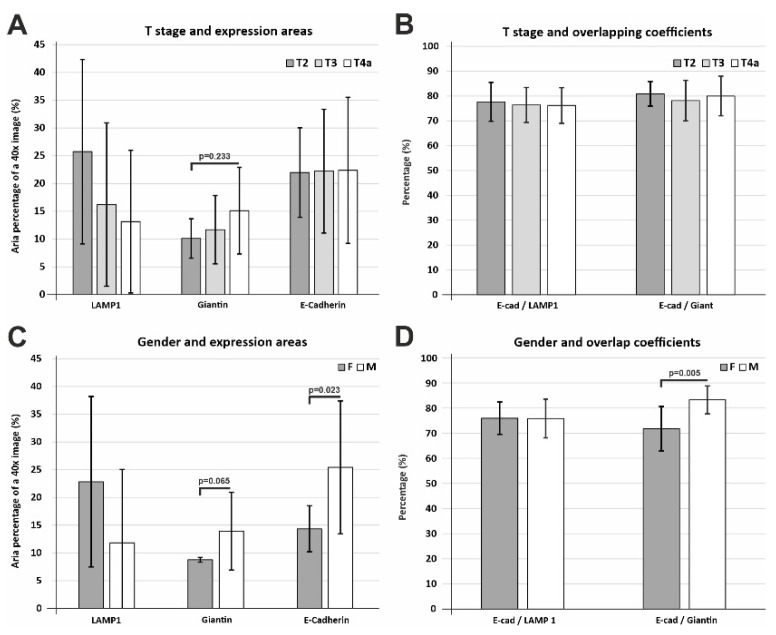
Analysis of T stage and gender denominators. (**A**) LAMP1 and Giantin expression areas show an inverse, albeit statistically non-significant, variation depending on the tumor stage. (**B**) E-cadherin colocalization with LAMP1 and Giantin show no differences when stratified for the three tumor stages. (**C**) Expression areas of the three markers showed that LAMP1 had a tendency for higher values in females, while Giantin and E-cadherin were decreased in females, with E-cadherin variations only showing a significant difference. (**D**) Overlapping coefficients for E-cadherin/LAMP1 do not show any variation depending on the gender of the patients, while E-cadherin/Giantin shows a clear-cut decrease for female patients. Significance is shown for a global ANOVA testing (**A**) and Student’s t-test (**C**,**D**). Data are expressed as means ± SD.

**Figure 10 biomedicines-10-00349-f010:**
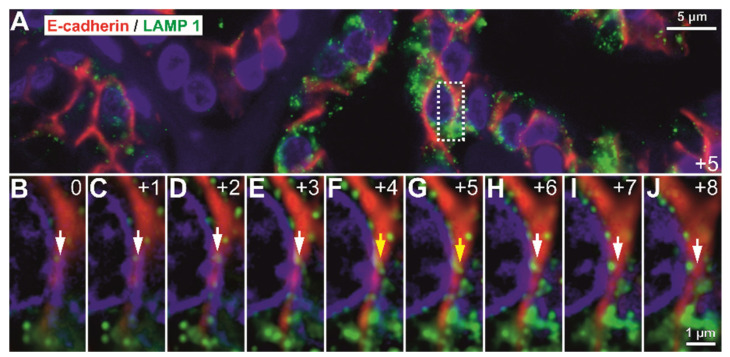
Fluorescence deconvolution optical z-stack sectioned at 0.1µm step. (**A**) An inset (rectangle) in the overall tumor tissue is followed in (**B**–**J** seriate optical sections and shows the E-cadherin stained cellular membranes with LAMP1-positive vesicles colocalizing with the membranes and the immediate sub-membrane staining, and follow-up with larger exosome/lysosome peri-membrane vesicles (white arrows) reveals continuous bodies of the cisternae traveling completely across the membrane from one cell to the other (yellow arrows). Numbers in Figures (**B**–**J**) represent optical sectioning planes starting from the surface of the tissue (0). Scale bar for image A, 5 µm; for images (**B**–**J**), 1 µm.

**Table 1 biomedicines-10-00349-t001:** Patients included in the study.

Patient No.	Degree of Differentiation	Age	Gender	Localization	Tumor Stage	Lymph Node Stage
1	Tubular, WD	72	F	Cardia	T3	N1
2	Tubular, MD	51	M	Body	T2	N2
3	Tubular, MD	54	M	Body	T2	N1
4	Tubular, MD	78	F	Body	T3	N3a
5	Tubular, WD	76	M	Pyloric	T3	N1
6	Tubular, MD	69	M	Body	T3	N2
7	Tubular, WD	65	F	Cardia	T4a	N3a
8	Tubular, MD	55	M	Body	T2	N2
9	Tubular, MD	69	M	Cardia	T3	N2
10	Solid, PD	69	M	Body	T4a	N1
11	Solid, PD	52	F	Body	T4a	N1
12	Solid, PD	65	M	Body	T4a	N2
13	Solid, PD	52	F	Body	T4a	N1
14	Solid, PD	44	F	Body	T3	N3b
15	Solid, PD	44	M	Pyloric	T3	N2
16	Solid, PD	78	M	Body	T3	N1
17	Solid, PD	79	M	Body	T4a	N2
18	Solid, PD	60	M	Body	T4a	N0
19	Control	67	F	-	-	-
20	Control	59	M	-	-	-
21	Control	78	M	-	-	-
22	Control	68	M	-	-	-
23	Control	75	F	-	-	-

^WD^ Well Differentiated, ^MD^ Moderately Differentiated, ^PD^ Poorly Differentiated.

## Data Availability

All the data presented in this study are available upon request from the corresponding author.

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
