# Peer review of "E-Cadherin Modulation and Inter-Cellular Trafficking in Tubular Gastric Adenocarcinoma: A High-Resolution Microscopy Pilot Study"

_biomedicines, 2022, doi:10.3390/biomedicines10020349_

Round 1
Reviewer 1 Report
The authors have satisfactorily addressed the issues raised previously.
Author Response
Re: We thank the Reviewer for all through previous issues that increased the quality of our presentation.
Reviewer 2 Report
In this manuscript, the authors have studied the mechanism of E-cadherin regulation in gastric adenocarcinoma. Using human tissues, an increased colocalization of E-cadherin with the Golgian vesicles and a decrease in the lysosomal pathways do not translate into a higher membrane expression with increasing grade of the tumors. This is an interesting study. However, the following points should be addressed,
- The novelty of the study should be clearly emphasized.
- High magnification images are needed for Figures 4, 5, and 6.
- The authors should increase the number of patient samples.
- The conclusions on the association between post-translational modification, vesicular trafficking, and E-cadherin expression should be confirmed by in vitro studies.
Author Response
Comments and Suggestions for Authors
X Moderate English changes required
X All sections of the manuscript can be improved
Re: We have thoroughly re-read the manuscript and made changes to better reflect its message.
In this manuscript, the authors have studied the mechanism of E-cadherin regulation in gastric adenocarcinoma. Using human tissues, an increased colocalization of E-cadherin with the Golgian vesicles and a decrease in the lysosomal pathways do not translate into a higher membrane expression with increasing grade of the tumors. This is an interesting study. However, the following points should be addressed,
Q1: The novelty of the study should be clearly emphasized.
Re1: Thank you very much for your suggestion. Taking the Reviewer’s suggestion, besides re-evaluating the whole manuscript, we have now clearly stated in the abstract that this study parallels previous data of in vitro experiments and histopathology showing that for partial Epithelial-to Mesenchymal Transition, E-cadherin is not in fact down-regulated, but rather redirected away from its normal processing pathway. And the strength of the paper is that it proves this concept on diagnostic pathological tissue characterized by polyclonal non-homogenous populations of tumor cells.
Q2: High magnification images are needed for Figures 4, 5, and 6.
Re2: Taking the Reviewer’s suggestion, we have increased the magnification in our main fluorescence images (ie Figures 4, 5, 6, and Supplementary Figure 2). Given the fact that we had captured large areas and high-resolution images, we still utilized our initial images as they were captured for the image analysis per se and thus represent exemplary working models. Moreover, all images associated with our manuscript have now been exported as 600 dpi TIFF CMYK files for best printing and visualization. We kindly ask the Reviewer to download and evaluate the native TIFF files, compression of the images in both word and pdf documents reduce image clarity and details.
Q3: The authors should increase the number of patient samples.
Re3: As discussed in the previous round of review, we have pointed out not only that the workload volume would not allow us to integrate more cases in a constant methodological approach (ie staining times for all IHC steps) considering the semi-quantitative nature of the paper. Moreover, a more in-depth study is feasible (here especially related to the observation of trans-membrane vesicle trafficking) only with state-of-the art equipment that we do not have yet available in our morphology imaging facility (ie confocal slide scanner and super—resolution microscopy). This is the reason why we kept the study as a proof-of principle or a pilot analysis, and altogether its importance resides in the fact that it offers the first view of these phenomena on routine diagnostic histopathology tissue and not on a more controlled in vitro setting.
Q4: The conclusions on the association between post-translational modification, vesicular trafficking, and E-cadherin expression should be confirmed by in vitro studies.
Re4: As pointed out in the previous round of review, our manuscript’s main finding is that it describes a discrepancy between the apparent net increase of E-cadherin production in the pre-membrane stages (increased E-Cadherin synthesis in the Golgian compartment and low recycling in the lysosomal compartment) and the drop in its membrane-associated expression, in cancer tissue; on FFPE routinely processed tissue for diagnostic purpose. It is essential to consider this last point, that this study was intended for routinely processed FFPE tissue, in order to get a valid interpretation from this point of view. Based on our data, we initially hypothesized that this discrepancy must be explained by subtle changes in maturation, post-translational modification and intracellular trafficking. However, at that time we were not aware of existing literature that indicate that tumor cells exist with a mosaicism of Epithelial-to Mesenchymal Transition (EMT) states, with cells with partial EMT (P-EMT) phenotypes alternating to complete EMT (C-EMT) [1, 2]. During P-EMT, E-cadherin is in fact not downregulated, but redistributed away from the membrane in recycling vesicles. We had included these data in the revised manuscript, as they made our conclusion much stronger, and again, reflect the validity of this mosaicism on FFPE tissue diagnostic grade tissue.
In fact, this phenomenon has been proved on in vitro studies [2], but indeed not for gastric adenocarcinoma cell lines, and taking the Reviewer’s suggestion, we have now clearly stated in the Discussion section that “future functional studies on gastric adenocarcinoma cell lines are clearly needed to shed light…” on these subtle synthesis chain changes. The expertise of our group is morphology-based image analysis at the tissue level, and thus our contribution to the field should remain the demonstration of the existence of E-cadherin processing pathway alteration in routinely diagnostic FFPE gastric adenocarcinoma. With more image analysis tools available, we will surely continue this work from the morphological point of view, an we would be more than open to any collaboration for extending the observation with in vivo experiments, but that would be with the expertise of a dedicated group with experience in this direction.
Altogether, we thank the Reviewer for strengthening the presentation impact and imaging support data of our paper.
References:
[1] Fang C, Kang Y. E-Cadherin: Context-Dependent Functions of a Quintessential Epithelial Marker in Metastasis. Cancer Res, 2021, 81(23):5800-5802. doi: 10.1158/0008-5472.CAN-21-3302. PMID: 34853039;
[2] Aiello NM, Maddipati R, Norgard RJ, Balli D, Li J, Yuan S, Yamazoe T, Black T, Sahmoud A, Furth EE, Bar-Sagi D, Stanger BZ. EMT Subtype Influences Epithelial Plasticity and Mode of Cell Migration. Dev Cell, 2018, 45(6):681-695 e684. doi: 10.1016/j.devcel.2018.05.027. PMID: 29920274; PMC6014628: PMC6014628.
Round 2
Reviewer 2 Report
The manuscript was improved by revision and can be accepted for publication.